# A Polarization-Insensitive Recirculating Delayed Self-Heterodyne Method for Sub-Kilohertz Laser Linewidth Measurement

Jing Gao [1,2,3], Dongdong Jiao [1,3], Xue Deng [1,3], Jie Liu [1,3], Linbo Zhang [1,2,3], Qi Zang [1,2,3], Xiang Zhang [1,2,3], Tao Liu [1,3,*] and Shougang Zhang [1,3]

1 National Time Service Center, Chinese Academy of Sciences, Xi'an 710600, China; gaojing@ntsc.ac.cn (J.G.); jiaodd@ntsc.ac.cn (D.J.); dengxue@ntsc.ac.cn (X.D.); jieliu2380@ntsc.ac.cn (J.L.); linbo@ntsc.ac.cn (L.Z.); zangqi@ntsc.ac.cn (Q.Z.); zhangxiang@ntsc.ac.cn (X.Z.); szhang@ntsc.ac.cn (S.Z.)
2 University of Chinese Academy of Sciences, Beijing 100039, China
3 Key Laboratory of Time and Frequency Standards, Chinese Academy of Sciences, Xi'an 710600, China
* Correspondence: taoliu@ntsc.ac.cn; Tel.: +86-29-8389-0519

**Abstract:** A polarization-insensitive recirculating delayed self-heterodyne method (PI-RDSHM) is proposed and demonstrated for the precise measurement of sub-kilohertz laser linewidths. By a unique combination of Faraday rotator mirrors (FRMs) in an interferometer, the polarization-induced fading is effectively reduced without any active polarization control. This passive polarization-insensitive operation is theoretically analyzed and experimentally verified. Benefited from the recirculating mechanism, a series of stable beat spectra with different delay times can be measured simultaneously without changing the length of delay fiber. Based on Voigt profile fitting of high-order beat spectra, the average Lorentzian linewidth of the laser is obtained. The PI-RDSHM has advantages of polarization insensitivity, high resolution, and less statistical error, providing an effective tool for accurate measurement of sub-kilohertz laser linewidth.

**Keywords:** the linewidth measurement of narrow laser; recirculating delayed loop; polarization insensitivity; self-heterodyne of beat spectrum

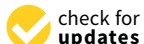



## 1. Introduction

With advantages of long coherence length and low phase noise, narrow-linewidth lasers are widely used in many fields, such as optical clocks, gravitational wave detection, coherent communications, and optical fiber sensors [1–4]. Laser linewidth has an important effect on characterizing the measurement range, precision, and sensitivity of those applications. Hence, accurate measurement of narrow laser linewidth is essential.

The conventional delayed self-heterodyne method (DSHM), which was first proposed by Okoshi, has been widely used to deduce the laser linewidth [5]. Compared with the heterodyne detection, an extra narrow-linewidth laser is not needed as the reference source. Usually, the Lorentzian linewidth of the self-heterodyne spectrum is used to characterize the laser linewidth, requiring the delay fiber of the DSHM must be at least five times longer than the coherence length of the laser under test [6,7]. Recently, the linewidths of commercial lasers have been decreased to the order of kilohertz, for which a delay fiber over hundreds of kilometers is requested by DSHM [8,9]. However, such a long fiber will cause high propagation loss, the polarization-induced fading, and the broadening spectrum, affecting accurate measurement of linewidth [10].

Some research teams have utilized improved DSHMs to measure the ultra-narrow laser linewidth [11–13]. X. P. Chen et al. demonstrated a sub-kilohertz laser linewidth measurement based on a loss-compensated recirculating delayed self-heterodyne interferometer (RDSHI) with 25 km delay fiber [14]. Due to the recirculating delayed loop with

a fiber amplifier, the linewidth resolution of RDSHI has improved to the order of sub-kilohertz laser. In 2020, Zhihui Wang et al. used the coherent envelope of self-heterodyne spectrum to estimate the laser linewidth by DSHM with short delay fiber, which can almost ignore the influence of the $1/f$ frequency noise [15]. However, the length of delay fiber has to be changed to find a suitable coherent envelope, which increases the complexity of the measurement. In those improved DSHMs, the polarization controller (PC) has been inserted into one arm of interferometer to match the polarization of the other arm. However, the state of polarization (SOP) may drift randomly due to the fiber birefringence effect and cannot be fully controlled by PC, resulting in the instability of the self-heterodyne spectrum and linewidth measurement error [16].

In this paper, we propose a PI-RDSHM for measuring the sub-kilohertz laser linewidth without any PC or changing the fiber length. The combination of three FRMs guarantees that the output SOPs of interferometer are unrelated to the fiber birefringence, providing a passive polarization-insensitive technique. By multiple passes in a 10 km fiber loop with an acoustic optical modulator (AOM), a series of stable beat spectra with different delay times are obtained simultaneously. To accurately measure the Lorentzian linewidth (natural linewidth of laser), the Voigt fitting is utilized to filter out the Gaussian spectrum which broadening the spectral linewidth. The statistical error of linewidth measurement is effectively decreased by averaging the Lorentzian linewidths of high-order beat notes. With the advantages of polarization insensitivity and less statistical error, the PI-RDSHM is one of the best candidates for precise measurement of sub-kilohertz laser linewidth.

## 2. Experiment Setup

The schematic setup of the laser linewidth measurement is shown in Figure 1. The laser source is a narrow-linewidth fiber laser. After passing through an optical isolator, the laser beam with a power of 2 mW enters into the improved DSHI. The input laser is first split into two paths by a 50/50 fiber coupler (C1). 50% of the input laser serving as the local reference beam enters into the short arm of this interferometer, which consists of a variable optical attenuator (VOA) and FRM1. The FRM that is composed of a 45 degrees Faraday rotator and a mirror reflects the input beam along the same path. Another part of the input laser serves as the signal beam and is injected into the long arm. After reflecting by FRM2, a large part of signal beam enters into the recirculating delay loop, which consists of an FRM2, C2, 10 km delay fiber spool (delay time 50 μs), an optical bandpass filter (OBF), an erbium-doped fiber amplifier (EDFA), AOM and FRM3. The OBF is used to suppress the amplified spontaneous emission noise of EDFA, which compensates for the transmission loss of the fiber loop. The AOM that has a center frequency of 40 MHz is used to shift the frequency of the circulating beam. The input power of PD is approximately 80 μW, and its bandwidth is 2 GHz. The combination of FRM2 and FRM3 is just like a fiber resonant cavity, so the signal beam can circulate in the fiber delay loop. After $n$ times circulations, the delay beam has a frequency shift of $2n \cdot 40$ MHz and a delay time of $2n \cdot 50$ μs. For every circulation, a small part of signal beam departs the recirculating delay loop from port 5 of C2, and finally beats with the local reference at C1. As a result, the PD detects self-heterodyne beat notes with frequencies of $2n \cdot 40$ MHz. It's well known that the linewidth resolution of DSHM is improved with the increase of delay time, so the resolution of the n-order beat spectrum for the PI-RDSHM is $2n$ times better than the conventional DSHM with the same length of fiber spool. Moreover, it is beneficial to reduce the statistical error by obtaining the average Lorentzian linewidth from several high-order spectra.

For a narrow-linewidth laser, the contribution of environmental noises to the fiber interferometer is not negligible, broadening the linewidth spectrum of laser [17]. To suppress the effect of environmental noises, the fiber spool is set in a cylindroid sealed aluminum can and the interferometer is placed inside a thick aluminum box of volume about 0.2 m$^3$. The box is covered by acoustic and thermal isolation foams, and finally set onto a vibration isolation platform.

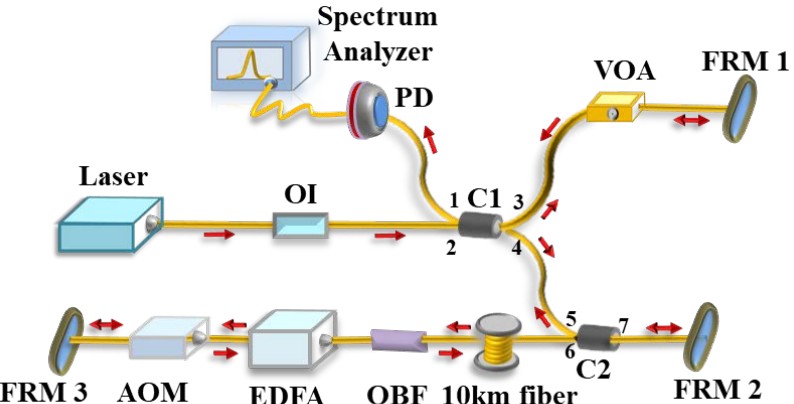

**Figure 1.** Schematic setup of laser linewidth measurement. OI: optical isolator, C: coupler, VOA: variable optical attenuator, OBF: optical bandpass filter, EDFA: erbium-doped fiber amplifier, AOM: acoustic optical modulator, FRM: Faraday rotator mirror, PD: photodetector.

## 3. Analysis of Polarization Insensitivity

Since the SOPs of interfering beams may drift randomly due to the fiber birefringence effect, the intensity of beat signal also fluctuates accordingly, resulting in polarization-induced fading [18]. PI-RDSHM is a passive polarization-insensitive operation for fiber recirculating interferometer. In this interferometer, the combination of FRMs guarantees that the output SOPs of interferometer are uncorrelated to the fiber birefringence effect. The evolution of SOPs for interfering beams are theoretically discussed in the following. To simplify the calculation, the transmission losses of the fiber devices and optical fiber are ignored.

The birefringence effect of ordinary single-mode fiber can be regarded as an elliptical retarder, so the forward transfer Jones matrix of fiber can be expressed as [19]

$$R_+ = \frac{1}{d} \begin{bmatrix} a & -b^* \\ b & a^* \end{bmatrix},$$ (1)

where $*$ denotes the conjugate, $d^2 = a \cdot a^* + b \cdot b^*$, and the elements $a$ and $b$ are related to the birefringence properties of the fiber. When the beam reflected by FRM along the same fiber path in the reverse direction, this single-mode fiber can be regarded as a reverse elliptical retarder, its reverse transfer Jones matrix can be written as

$$R_- = \frac{1}{d} \begin{bmatrix} a & -b \\ b^* & a^* \end{bmatrix},$$ (2)

The FRM that consists of a 45 degrees Faraday rotator and retro-reflector mirror is expressed by Jones matrix as

$$T_F = \begin{bmatrix} 0 & -1 \\ -1 & 0 \end{bmatrix},$$ (3)

The Jones matrix of direct coupling and the cross-coupling of C1 can be given as

$$J_D = J_{24} = J_{31} = \begin{bmatrix} \sqrt{A} & 0 \\ 0 & \sqrt{1-A} \end{bmatrix}, J_T = J_{23} = J_{41} = \begin{bmatrix} i\sqrt{A} & 0 \\ 0 & i\sqrt{1-A} \end{bmatrix},$$ (4)

where $A$ denotes the coupling coefficient of C1. The $J_{mn}$ represents the beam of fiber coupler from the input port of $m$ to the output port of $n$. The Jones matrix of direct coupling and the cross-coupling of C2 can be written as

$$J_t = J_{75} = J_{57} = \begin{bmatrix} i\sqrt{\alpha} & 0 \\ 0 & i\sqrt{1-\alpha} \end{bmatrix}, J_d = J_{67} = J_{76} = \begin{bmatrix} \sqrt{\alpha} & 0 \\ 0 & \sqrt{1-\alpha} \end{bmatrix}, \quad (5)$$

where $\alpha$ denotes the coupling coefficient of C2. According to the above matrixes, the evolution of SOP for local reference beam that transmits in the short arm is described by Jones matrix as

$$P_{out1} = J_{31} \cdot R_- \cdot T_F \cdot R_+ \cdot J_{23} = -J_{31} \cdot \begin{bmatrix} 0 & 1 \\ 1 & 0 \end{bmatrix} \cdot J_{23} = -i\sqrt{A(1-A)} \cdot \begin{bmatrix} 0 & 1 \\ 1 & 0 \end{bmatrix}, \quad (6)$$

The Equation (6) indicates that the output SOP of the short arm is independent of the fiber birefringence effect due to the constant components of $P_{out1}$ [20]. In other words, the output SOP of the short arm is only related to the input SOP. After n circulations, the evolution of SOP for the long arm can be calculated as

$$\begin{aligned} P_{out2} &= J_{41}R_-J_{75} \cdot (R_-T_FR_+J_{67}R_-T_FR_+J_{76})^n \cdot R_-T_FR_+J_{57}R_+J_{24} \\ &= [\sqrt{\alpha(1-\alpha)}]^n \cdot J_{41}R_-J_{75} \cdot \begin{bmatrix} 1 & 0 \\ 0 & 1 \end{bmatrix}^n \cdot R_-T_FR_+J_{57}R_+J_{24} \\ &= [\sqrt{\alpha(1-\alpha)}]^{n+1} \cdot i\sqrt{A(1-A)}) \cdot \begin{bmatrix} 0 & 1 \\ 1 & 0 \end{bmatrix} \end{aligned} \quad (7)$$

It's worth noting that the evolution of SOP for the recirculating delay loop is a unit matrix. No matter how many times the signal beam transmits in the delay loop, the output SOP and the input SOP of the delay loop at the 7 port is identical to each other. The Equation (7) showed that the evolution of SOP for the long arm is also independent of the fiber birefringence effect due to the constant components of $P_{out2}$. The above equations enable us to conclude that the combination of FRMs eliminates the random variety of polarization without any PC. Consequently, the output SOPs of the two interfering beams are consistent with each other.

Meanwhile, the polarization insensitivity of this PI-RDSHM is investigated experimentally. The intensity of beat signal is directly related to the output SOPs of the interfering beams, so the intensity variation of beat spectra with fiber birefringence can be used to deduce whether this interferometer is polarization-insensitive. For testing the polarization sensitivity of fiber interferometer, a 3-paddle PC that utilizes stress-induced birefringence to alter the SOP of the delay loop fiber is placed between the 10 km fiber and C2. Figure 2 shows intensity variations of high-order beat spectra with random SOP adjustment in the normal RDSHI, where the FRM3 of PI-RDSHM is replaced by a retro-reflector mirror. The orange line represents a spectrum of the tenth-order beat note (S10) and the blue line indicates a spectrum of the eleventh-order beat note (S11). As the Figure 2 shown, there are obvious intensity variations of S10 and S11 with rotating Paddles of PC. The maximum intensity difference of S10 is approximately 16.7 dB. Figure 3 shows the effect of polarization variation on S10 and S11 by PI-RDSHM. The intensity variation of S10 and S11 are both lower than 1 dB. A comparison of Figure 2 with Figure 3 shows that the PI-RDSHM can effectively suppress the polarization-induced fading and provide stable self-heterodyne spectra.

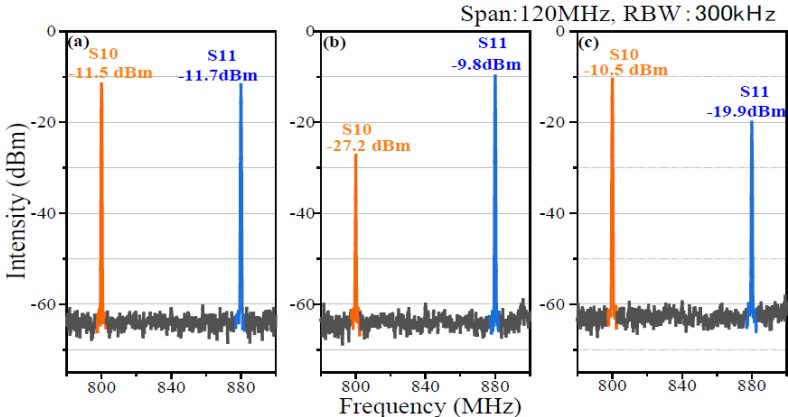

**Figure 2.** Typical intensity variations of S10 and S11 with random SOP adjustment for normal RDSHI. (**a**) The spectra of S10 and S11 obtaind from the RDSHI; (**b,c**) the spectra of S10 and S11 with random SOP adjustment without FRMs in system.

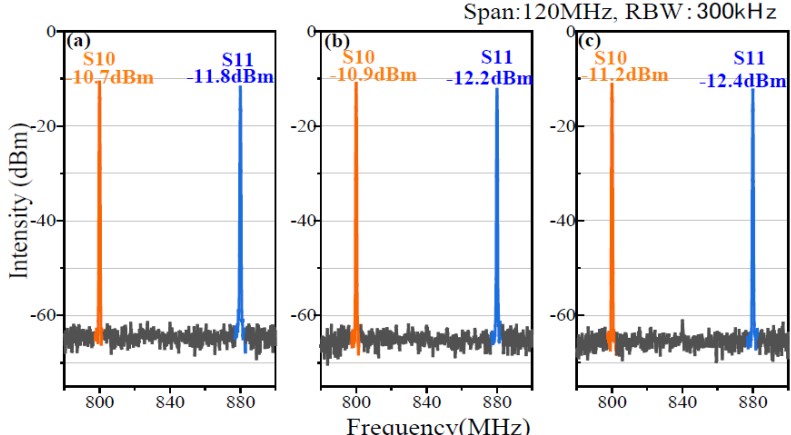

**Figure 3.** Typical intensity variation of S10 and S11 with random SOP adjustment for polarization-insensitive RDSHI. (**a**) The spectra of S10 and S11 obtain from the polarization-insensitive RDSHI; (**b,c**) the spectra of S10 and S11 with random SOP adjustment with FRMs in system.

## 4. Experiment Results

As shown in Figure 4, a series of beat notes are recorded by a spectrum analyzer with the resolution bandwidth (RBW) of 1 MHz. There are 20 orders of beat spectra in the range of 1.6 GHz with an interval of 80 MHz. The S12 and S13 indicate spectra of the twelfth and thirteenth-order beat notes respectively.

The equivalent delay line of S10 is 200 times longer than the 10 km fiber, which is long enough for the resolution of sub-kilohertz laser linewidth measurement [21]. Figure 5 shows the measured spectrum and the fitted curves of S10, where the green dash-dotted line represents the Lorentzian curve fitting and the red line indicates the Voigt curve fitting. Apparently, the red Voigt profile agrees well with the measured spectrum of S10 that is shown by grey dots, while the Lorentzian profile deviates significantly in the center part of measured data of the S10. In fact, the laser frequency noise includes the white frequency noise (spontaneous emission of laser) and the $1/f$ noise, so the self-heterodyne spectrum of DSHM contains two components. One component is a Lorentzian spectrum (natural linewidth) caused by the white frequency noise. The other is an approximate Gaussian spectrum associated with the $1/f$ noise [22,23]. The Voigt profile is the convolution of the Lorentzian spectrum and the approximately Gaussian spectrum, so the Voigt profile can satisfactorily interpret the detected spectrum of the PI-RDSHM.

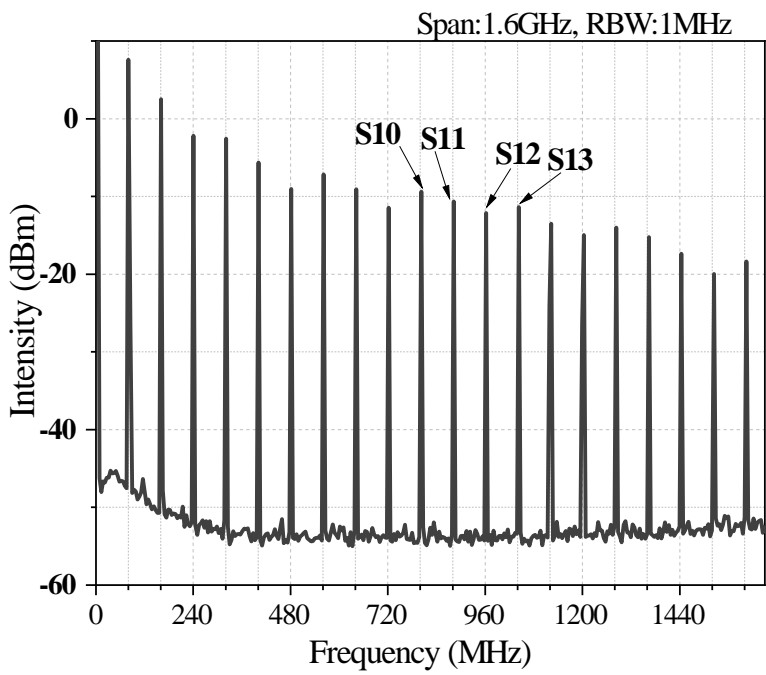

**Figure 4.** Self-heterodyne beat spectra of the PI-RDSHM.

By Voigt profile fitting, the Lorentzian linewidth and the Gaussian linewidth can be well distinguished [24]. We analyze several high-orders spectra and the linewidth results by Voigt profile fitting are shown in Figure 6. The green dots indicate the Lorentzian linewidths and the black triangles represent the Gaussian linewidths of high-orders beat notes (from 9th to 13th order). The green and black dash lines indicated the mean value of Lorentzian and Gaussian linewidth, which is 0.86 kHz and 2.74 kHz respectively. The error bars and shadow areas represent the fitting errors and standard deviations. It is obvious that the Gaussian linewidths are broadening with the increase of the length of delay fiber, while Lorentzian linewidths have small change. For sub-kilohertz laser, the Lorentzian linewidth would be drowned by the Gaussian linewidth, so the Voigt fitting method that separate the Lorentzian spectrum from Gaussian spectrum is necessary to estimate the natural linewidth. By averaging the Lorentzian linewidths of high-order beat notes, the statistical error of measurement is reduced effectively.

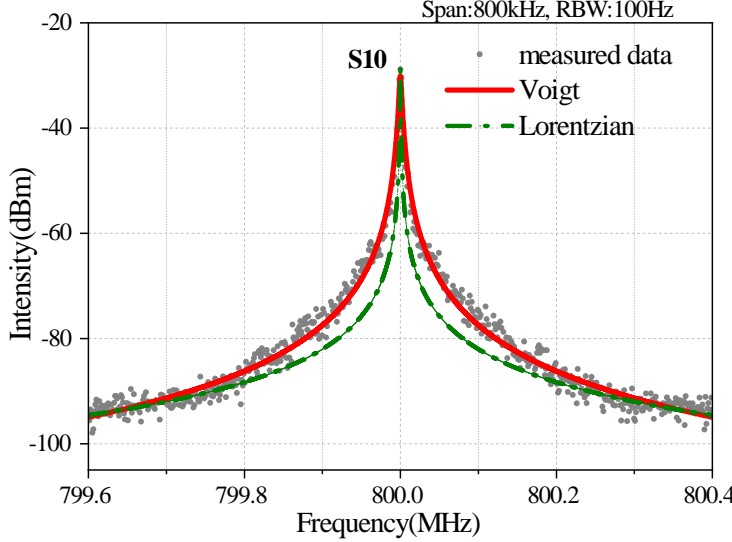

**Figure 5.** The S10 with the 200 km equivalent delay fiber and its fitting curves.

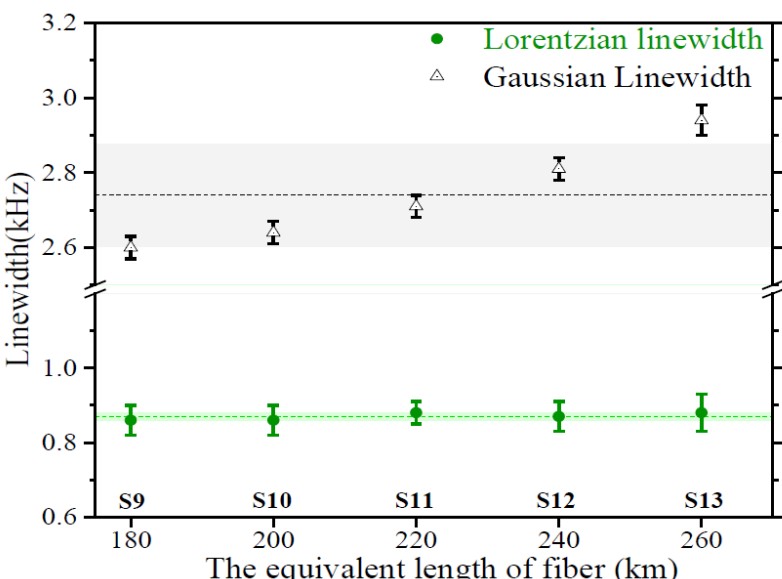

**Figure 6.** The Lorentzian linewidths and Gaussian linewidths of high-orders beat notes by Voigt profile method.

## 5. Conclusions

In summary, we present PI-RDSHM with only a 10 km fiber spool for sub-kilohertz laser linewidth measurement without any PC. The combination of FRMs guarantees that the output SOPs of interferometer are uncorrelated to the fiber birefringence effect, which has been theoretically analyzed through the Jones matrixes and experimentally verified. Due to the recirculation of delayed beam, the resolution of n-order beat notes is $2n$ times better than the conventional DSHM with the same length of fiber spool, so high-orders beat notes can obtain enough delay time for the resolution of sub-kilohertz linewidth measurement. By analyzing several high-orders spectra, the average Lorentzian linewidth of laser is obtained. The PI-RDSHM can effectively suppress the polarization-induced fading, improve the linewidth resolution and decrease the statistical error, so it provides a powerful candidate for precise measurement of sub-kilohertz laser linewidth.

**Author Contributions:** Conceptualization, T.L. and S.Z.; methodology, J.G.; validation, J.G., D.J. and J.L.; formal analysis, J.G.; data curation, Q.Z. and X.Z.; writing—original draft preparation, J.G.; writing—review and editing, J.G., X.D. and L.Z.; supervision, T.L. All authors have read and agreed to the published version of the manuscript.

**Funding:** This research was funded by the National Natural Science Foundation of China (Grant Nos. 91636101, 91836301, and 11803041), the West Light Foundation of the Chinese Academy of Sciences (Grant No. XAB2016B47), and the Strategic Priority Research Program of the Chinese Academy of Sciences (Grant No. XDB21000000).

**Conflicts of Interest:** The authors declare that there is no conflict of interest.

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
