# Peer review of "A Polarization-Insensitive Recirculating Delayed Self-Heterodyne Method for Sub-Kilohertz Laser Linewidth Measurement"

_photonics, doi:10.3390/photonics8050137_

Round 1
Reviewer 1 Report
Line 57: optical isolator and not optical isolate
Author Response
Thank you very much for your reviews and comments to our manuscript (manuscript ID: photonics-1163770, Type: Communication, Title: A polarization-insensitive recirculating delayed self-heterodyne method for sub-kilohertz laser linewidth measurement). We have carefully revised the manuscript according to the comments.

Reviewer 2 Report
The manuscript by J. Gao, et al. describes a polarization-insensitive recirculating delayed self-heterodyne method for precision measurement of
sub-kHz linewidths of a fiber laser. The method was demonstrated clearly by the detailed calculation and experimental results. Before the consideration of publications, I have a few comments that need to be addressed.
- In the introduction, the author mentioned the improved delayed self-heterodyne method (DSHM) for narrow-linewidth measurements, such as the recirculating-DSHM used in the paper. This conception and method have been proposed and investigated very early by H. Tsuchida. Tsuchida H. Simple technique for improving the resolution of the delayed self-heterodyne method[J]. Optics letters, 1990, 15(11): 640-642.
- The experimental setup shown in Fig. 1 is not intuitive enough. Could the author add the arrows to depict the light propagation direction?
- In Fig. 2 and Fig. 3, it is better to add some gab between (a) and (b), (b)and (c) to avoid any misunderstanding by the readers.
- When the authors present the linewidth results, is it for 3-dB or 20-dB linewidth? It should be clarified.
- The authors compared the Lorentzian linewidths and Gaussian linewidths for high-order beat notes (S10-S13). Could the author also add the results of low-order beat notes S1-S9? I presume that the low-order (maybe S1 and S2) can not give the precision measurement results due to the limited delay length. However, more data in Fig. 6 will help to support their theoretical analysis.
- In line 57, "isolate" should be "isolator"
- In line 58, "mw" should be "mW"
Author Response

(The authors gave the same response as above.)

Reviewer 3 Report
The paper presents a polarization-insensitive recirculating delayed self-heterodyne method (PI-RDSHM) used to measure the linewidth of sub-kilohertz lasers. Conventional DSHM systems have the disadvantage of requiring polarization controller to control the state of polarization otherwise the spectrum SNR could degrade in time. This work propose a combination of three Faraday rotator mirrors to replace the polarization controller and have passive polarization insensitive technique.
The experimental results show that the 10th and 11th order beat-notes are independent of the polarization and linewidth measurement by fitting with a Voigt function is presented.
At Line 128 probably there is a typo regarding the word “with”
Author Response

(The authors gave the same response as above.)
